# ELF: An Extensive, Lightweight and Flexible Research Platform for Real-time Strategy Games

**Yuandong Tian**[1]    **Qucheng Gong**[1]    **Wenling Shang**[2]    **Yuxin Wu**[1]    **C. Lawrence Zitnick**[1]

[1]Facebook AI Research        [2]Oculus

[1]{yuandong, qucheng, yuxinwu, zitnick}@fb.com    [2]wendy.shang@oculus.com

## Abstract

In this paper, we propose *ELF*, an Extensive, Lightweight and Flexible platform for fundamental reinforcement learning research. Using ELF, we implement a highly customizable real-time strategy (RTS) engine with three game environments (Mini-RTS, Capture the Flag and Tower Defense). Mini-RTS, as a miniature version of StarCraft, captures key game dynamics and runs at 40K frame-per-second (FPS) per core on a laptop. When coupled with modern reinforcement learning methods, the system can train a full-game bot against built-in AIs end-to-end in one day with 6 CPUs and 1 GPU. In addition, our platform is flexible in terms of environment-agent communication topologies, choices of RL methods, changes in game parameters, and can host existing C/C++-based game environments like ALE [4]. Using ELF, we thoroughly explore training parameters and show that a network with Leaky ReLU [17] and Batch Normalization [11] coupled with long-horizon training and progressive curriculum beats the rule-based built-in AI more than $70\%$ of the time in the full game of Mini-RTS. Strong performance is also achieved on the other two games. In game replays, we show our agents learn interesting strategies. ELF, along with its RL platform, is open sourced at `https://github.com/facebookresearch/ELF`.

## 1 Introduction

Game environments are commonly used for research in Reinforcement Learning (RL), i.e. how to train intelligent agents to behave properly from sparse rewards [4, 6, 5, 14, 29]. Compared to the real world, game environments offer an infinite amount of highly controllable, fully reproducible, and automatically labeled data. Ideally, a game environment for fundamental RL research is:

- **Extensive**: The environment should capture many diverse aspects of the real world, such as rich dynamics, partial information, delayed/long-term rewards, concurrent actions with different granularity, etc. Having an extensive set of features and properties increases the potential for trained agents to generalize to diverse real-world scenarios.

- **Lightweight**: A platform should be fast and capable of generating samples hundreds or thousands of times faster than real-time with minimal computational resources (e.g., a single machine). Lightweight and efficient platforms help accelerate academic research of RL algorithms, particularly for methods which are heavily data-dependent.

- **Flexible**: A platform that is easily customizable at different levels, including rich choices of environment content, easy manipulation of game parameters, accessibility of internal variables, and flexibility of training architectures. All are important for fast exploration of different algorithms. For example, changing environment parameters [35], as well as using internal data [15, 19] have been shown to substantially accelerate training.

To our knowledge, no current game platforms satisfy all criteria. Modern commercial games (e.g., StarCraft I/II, GTA V) are extremely realistic, but are not customizable and require significant resources for complex visual effects and for computational costs related to platform-shifting (e.g., a virtual machine to host Windows-only SC I on Linux). Old games and their wrappers [4, 6, 5, 14]) are substantially faster, but are less realistic with limited customizability. On the other hand, games designed for research purpose (e.g., MazeBase [29], $\mu$RTS [23]) are efficient and highly customizable, but are not very extensive in their capabilities. Furthermore, none of the environments consider simulation concurrency, and thus have limited flexibility when different training architectures are applied. For instance, the interplay between RL methods and environments during training is often limited to providing simplistic interfaces (e.g., one interface for one game) in scripting languages like Python.

In this paper, we propose *ELF*, a research-oriented platform that offers games with diverse properties, efficient simulation, and highly customizable environment settings. The platform allows for both game parameter changes and new game additions. The training of RL methods is deeply and flexibly integrated into the environment, with an emphasis on concurrent simulations. On ELF, we build a real-time strategy (RTS) game engine that includes three initial environments including Mini-RTS, Capture the Flag and Tower Defense. Mini-RTS is a miniature custom-made RTS game that captures all the basic dynamics of StarCraft (fog-of-war, resource gathering, troop building, defense/attack with troops, etc). Mini-RTS runs at 165K FPS on a 4 core laptop, which is faster than existing environments by an order of magnitude. This enables us for the first time to train end-to-end a full-game bot against built-in AIs. Moreover, training is accomplished in only one day using 6 CPUs and 1 GPU. The other two games can be trained with similar (or higher) efficiency.

Many real-world scenarios and complex games (e.g. StarCraft) are hierarchical in nature. Our RTS engine has full access to the game data and has a built-in hierarchical command system, which allows training at any level of the command hierarchy. As we demonstrate, this allows us to train a full-game bot that acts on the top-level strategy in the hierarchy while lower-level commands are handled using build-in tactics. Previously, most research on RTS games focused only on lower-level scenarios such as tactical battles [34, 25]. The full access to the game data also allows for supervised training with small-scale internal data.

ELF is resilient to changes in the topology of the environment-actor communication used for training, thanks to its hybrid C++/Python framework. These include one-to-one, many-to-one and one-to-many mappings. In contrast, existing environments (e.g., OpenAI Gym [6] and Universe [33]) wrap one game in one Python interface, which makes it cumbersome to change topologies. Parallelism is implemented in C++, which is essential for simulation acceleration. Finally, ELF is capable of hosting any existing game written in C/C++, including Atari games (e.g., ALE [4]), board games (e.g. Chess and Go [32]), physics engines (e.g., Bullet [10]), etc, by writing a simple adaptor.

Equipped with a flexible RL backend powered by PyTorch, we experiment with numerous baselines, and highlight effective techniques used in training. We show the first demonstration of end-to-end trained AIs for real-time strategy games with partial information. We use the Asynchronous Advantagous Actor-Critic (A3C) model [21] and explore extensive design choices including frame-skip, temporal horizon, network structure, curriculum training, etc. We show that a network with Leaky ReLU [17] and Batch Normalization [11] coupled with long-horizon training and progressive curriculum beats the rule-based built-in AI more than $70\%$ of the time in full-game Mini-RTS. We also show stronger performance in others games. ELF and its RL platform, is open-sourced at `https://github.com/facebookresearch/ELF`.

## 2   Architecture

ELF follows a canonical and simple producer-consumer paradigm (Fig. 1). The producer plays $N$ games, each in a single C++ thread. When a batch of $M$ current game states are ready ($M < N$), the corresponding games are blocked and the batch are sent to the Python side via the daemon. The consumers (e.g., actor, optimizer, etc) get batched experience with history information via a Python/C++ interface and send back the replies to the blocked batch of the games, which are waiting for the next action and/or values, so that they can proceed. For simplicity, the producer and consumers are in the same process. However, they can also live in different processes, or even on different machines. Before the training (or evaluation) starts, different consumers register themselves for batches with

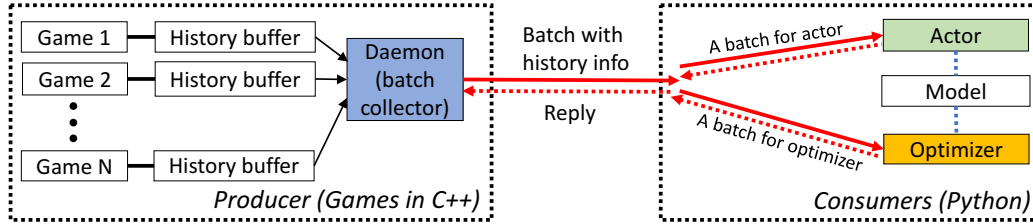

Figure 1: Overview of ELF.

different history length. For example, an actor might need a batch with short history, while an optimizer (e.g., $T$-step actor-critic) needs a batch with longer history. During training, the consumers use the batch in various ways. For example, the actor takes the batch and returns the probabilties of actions (and values), then the actions are sampled from the distribution and sent back. The batch received by the optimizer already contains the sampled actions from the previous steps, and can be used to drive reinforcement learning algorithms such as A3C. Here is a sample usage of ELF:

```
1   # We run 1024 games concurrently .
2   num_games = 1024
3
4   # Wait for a batch of 256 games.
5   batchsize = 256
6
7   # The return states contain key 's', 'r' and 'terminal'
8   # The reply contains key 'a' to be filled from the Python side .
9   # The definitions of the keys are in the wrapper of the game.
10  input_spec = dict (s='', r='', terminal='')
11  reply_spec = dict (a='')
12
13  context = Init (num_games, batchsize, input_spec, reply_spec )
```

Initialization of ELF

```
1   # Start all game threads and enter main loop.
2   context . Start ()
3   while True:
4       # Wait for a batch of game states to be ready
5       # These games will be blocked, waiting for replies .
6       batch = context .Wait()
7
8       # Apply a model to the game state . The output has key 'pi'
9       output = model(batch)
10
11      # Sample from the output to get the actions of this batch .
12      reply ['a' ][:] = SampleFromDistribution(output )
13
14      # Resume games.
15      context . Steps ()
16
17  # Stop all game threads .
18  context .Stop()
```

Main loop of ELF

**Parallelism using C++ threads.** Modern reinforcement learning methods often require heavy parallelism to obtain diverse experiences [21, 22]. Most existing RL environments (OpenAI Gym [6] and Universe [33], RLE [5], Atari [4], Doom [14]) provide Python interfaces which wrap only single game instances. As a result, parallelism needs to be built in Python when applying modern RL methods. However, thread-level parallelism in Python can only poorly utilize multi-core processors, due to the Global Interpreter Lock (GIL)[1]. Process-level parallelism will also introduce extra data exchange overhead between processes and increase complexity to framework design. In contrast, our parallelism is achieved with C++ threads for better scaling on multi-core CPUs.

**Flexible Environment-Model Configurations.** In ELF, one or multiple consumers can be used. Each consumer knows the game environment identities of samples from received batches, and typically contains one neural network model. The models of different consumers may or may not share parameters, might update the weights, might reside in different processes or even on different machines. This architecture offers flexibility for switching topologies between game environments and models. We can assign one model to each game environment, or *one-to-one* (e.g, vanilla A3C [21]), in which each agent follows and updates its own copy of the model. Similarly, multiple environments can be assigned to a single model, or *many-to-one* (e.g., BatchA3C [35] or GA3C [1]), where the model can perform batched forward prediction to better utilize GPUs. We have also incorporated forward-planning methods (e.g., Monte-Carlo Tree Search (MCTS) [7, 32, 27]) and Self-Play [27], in which a single environment might emit multiple states processed by multiple models, or *one-to-many*. Using ELF, these training configurations can be tested with minimal changes.

**Highly customizable and unified interface.** Games implemented with our RTS engine can be trained using raw pixel data or lower-dimensional internal game data. Using internal game data is

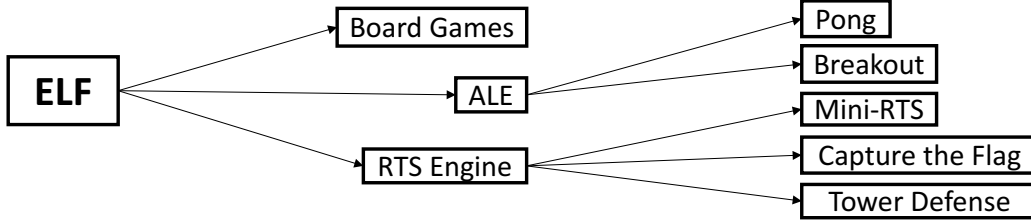

Figure 2: Hierarchical layout of ELF. In the current repository (https://github.com/facebookresearch/ELF, master branch), there are board games (e.g., Go [32]), Atari learning environment [4], and a customized RTS engine that contains three simple games.

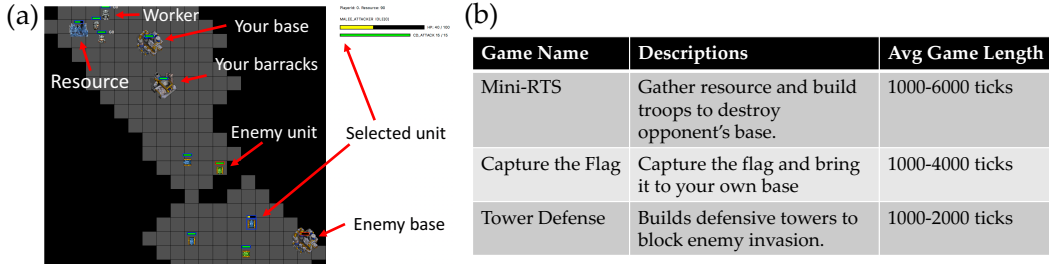

Figure 3: Overview of Real-time strategy engine. **(a)** Visualization of current game state. **(b)** The three different game environments and their descriptions.

typically more convenient for research focusing on reasoning tasks rather than perceptual ones. Note that web-based visual renderings is also supported (e.g., Fig. 3(a)) for case-by-case debugging.

ELF allows for a unified interface capable of hosting any existing game written in C/C++, including Atari games (e.g., ALE [4]), board games (e.g. Go [32]), and a customized RTS engine, with a simple adaptor (Fig. 2). This enables easy multi-threaded training and evaluation using existing RL methods. Besides, we also provide three concrete simple games based on RTS engine (Sec. 3).

**Reinforcement Learning backend.** We propose a Python-based RL backend. It has a flexible design that decouples RL methods from models. Multiple baseline methods (e.g., A3C [21], Policy Gradient [30], Q-learning [20], Trust Region Policy Optimization [26], etc) are implemented, mostly with very few lines of Python codes.

## 3 Real-time strategy Games

Real-time strategy (RTS) games are considered to be one of the next grand AI challenges after Chess and Go [27]. In RTS games, players commonly gather resources, build units (facilities, troops, etc), and explore the environment in the fog-of-war (i.e., regions outside the sight of units are invisible) to invade/defend the enemy, until one player wins. RTS games are known for their exponential and changing action space (e.g., $5^{10}$ possible actions for 10 units with 5 choices each, and units of each player can be built/destroyed when game advances), subtle game situations, incomplete information due to limited sight and long-delayed rewards. Typically professional players take 200-300 actions per minute, and the game lasts for 20-30 minutes.

Very few existing RTS engines can be used directly for research. Commercial RTS games (e.g., StarCraft I/II) have sophisticated dynamics, interactions and graphics. The game play strategies have been long proven to be complex. Moreover, they are close-source with unknown internal states, and cannot be easily utilized for research. Open-source RTS games like Spring [12], OpenRA [24] and Warzone 2100 [28] focus on complex graphics and effects, convenient user interface, stable network play, flexible map editors and plug-and-play mods (i.e., game extensions). Most of them use rule-based AIs, do not intend to run faster than real-time, and offer no straightforward interface with modern machine learning architectures. ORTS [8], BattleCode [2] and RoboCup Simulation League [16] are designed for coding competitions and focused on rule-based AIs. Research-oriented platforms (e.g., $\mu$RTS [23], MazeBase [29]) are fast and simple, often coming with various baselines,

| | Realistic | Code | Resource | Rule AIs | Data AIs | RL backend |
|---|---|---|---|---|---|---|
| StarCraft I/II | High | No | High | Yes | No | No |
| TorchCraft | High | Yes | High | Yes | Yes | No |
| ORTS, BattleCode | Mid | Yes | Low | Yes | No | No |
| $\mu$RTS, MazeBase | Low | Yes | Low | Yes | Yes | No |
| Mini-RTS | Mid | Yes | Low | Yes | Yes | Yes |

Table 1: Comparison between different RTS engines.

| Platform | ALE [4] | RLE [5] | Universe [33] | Malmo [13] |
|---|---|---|---|---|
| Frame per second | 6000 | 530 | 60 | 120 |
| Platform | DeepMind Lab [3] | VizDoom [14] | TorchCraft [31] | Mini-RTS |
| Frame per second | 287(C)/866(G) | $\sim 7,000$ | 2,000 (frameskip=50) | 40,000 |

Table 2: Frame rate comparison. Note that Mini-RTS does not render frames, but save game information into a C structure which is used in Python without copying. For DeepMind Lab, FPS is 287 (CPU) and 866 (GPU) on single 6CPU+1GPU machine. Other numbers are in 1CPU core.

but often with much simpler dynamics than RTS games. Recently, TorchCraft [31] provides APIs for StarCraft I to access its internal game states. However, due to platform incompatibility, one docker is used to host one StarCraft engine, and is resource-consuming. Tbl. 1 summarizes the difference.

## 3.1 Our approach

Many popular RTS games and its variants (e.g., StarCraft, DoTA, Leagues of Legends, Tower Defense) share the same structure: a few units are controlled by a player, to move, attack, gather or cast special spells, to influence their own or an enemy's army. With our command hierarchy, a new game can be created by changing (1) available commands (2) available units, and (3) how each unit emits commands triggered by certain scenarios. For this, we offer simple yet effective tools. Researchers can change these variables either by adding commands in C++, or by writing game scripts (e.g., Lua). All derived games share the mechanism of hierarchical commands, replay, etc. Rule-based AIs can also be extended similarly. We provide the following three games: Mini-RTS, Capture the Flag and Tower Defense (Fig. 3(b)). These games share the following properties:

**Gameplay.** Units in each game move with real coordinates, have dimensions and collision checks, and perform durative actions. The RTS engine is tick-driven. At each tick, AIs make decisions by sending commands to units based on observed information. Then commands are executed, the game's state changes, and the game continues. Despite a fair complicated game mechanism, Mini-RTS is able to run 40K frames-per-second per core on a laptop, an order of magnitude faster than most existing environments. Therefore, bots can be trained in a day on a single machine.

**Built-in hierarchical command levels.** An agent could issue strategic commands (e.g., more aggressive expansion), tactical commands (e.g., hit and run), or micro-command (e.g., move a particular unit backward to avoid damage). Ideally strong agents master all levels; in practice, they may focus on *a certain level* of command hierarchy, and leave others to be covered by hard-coded rules. For this, our RTS engine uses a hierarchical command system that offers different levels of controls over the game. A high-level command may affect all units, by issuing low-level commands. A low-level, unit-specific *durative* command lasts a few ticks until completion during which per-tick *immediate* commands are issued.

**Built-in rule-based AIs.** We have designed rule-based AIs along with the environment. These AIs have access to all the information of the map and follow fixed strategies (e.g., build 5 tanks and attack the opponent base). These AIs act by sending high-level commands which are then translated to low-level ones and then executed.

With ELF, for the first time, we are able to train full-game bots for real-time strategy games and achieve stronger performance than built-in rule-based AIs. In contrast, existing RTS AIs are either rule-based or focused on tactics (e.g., 5 units vs. 5 units). We run experiments on the three games to justify the usability of our platform.

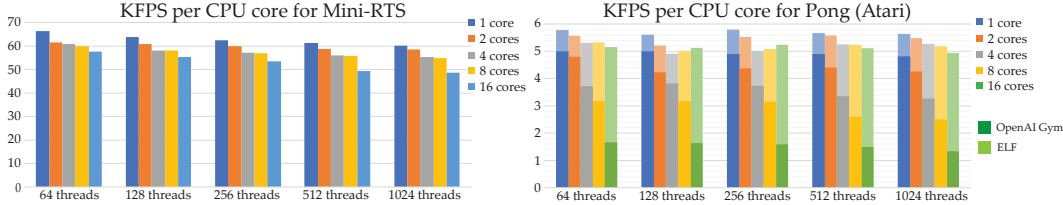

Figure 4: Frame-per-second per CPU core (no hyper-threading) with respect to CPUs/threads. ELF (light-shaded) is 3x faster than OpenAI Gym [6] (dark-shaded) with 1024 threads. CPU involved in testing: Intel E5-2680@2.50GHz.

# 4 Experiments

## 4.1 Benchmarking ELF

We run ELF on a single server with a different number of CPU cores to test the efficiency of parallelism. Fig. 4(a) shows the results when running Mini-RTS. We can see that ELF scales well with the number of CPU cores used to run the environments. We also embed Atari emulator [4] into our platform and check the speed difference between a single-threaded ALE and paralleled ALE per core (Fig. 4(b)). While a single-threaded engine gives around 5.8K FPS on Pong, our paralleled ALE runs comparable speed (5.1K FPS per core) with up to 16 cores, while OpenAI Gym (with Python threads) runs 3x slower (1.7K FPS per core) with 16 cores 1024 threads, and degrades with more cores. Number of threads matters for training since they determine how diverse the experiences could be, with the same number of CPUs. Apart from this, we observed that Python multiprocessing with Gym is even slower, due to heavy communication of game frames among processes. Note that we used no hyperthreading for all experiments.

## 4.2 Baselines on Real-time Strategy Games

We focus on 1-vs-1 full games between trained AIs and built-in AIs. Built-in AIs have access to full information (e.g., number of opponent's tanks), while trained AIs know partial information in the fog of war, i.e., game environment within the sight of its own units. There are exceptions: in Mini-RTS, the location of the opponent's base is known so that the trained AI can attack; in Capture the Flag, the flag location is known to all; Tower Defense is a game of complete information.

**Details of Built-in AI.** For Mini-RTS there are two rule-based AIs: SIMPLE gathers, builds five tanks and then attacks the opponent base. HIT_N_RUN often harasses, builds and attacks. For Capture the Flag, we have one built-in AI. For Tower Defense (TD), no AI is needed. We tested our built-in AIs against a human player and find they are strong in combat but exploitable. For example, SIMPLE is vulnerable to hit-and-run style harass. As a result, a human player has a win rate of 90% and 50% against SIMPLE and HIT_N_RUN, respectively, in 20 games.

**Action Space**. For simplicity, we use 9 strategic (and thus global) actions with hard-coded execution details. For example, AI may issue BUILD_BARRACKS, which automatically picks a worker to build barracks at an empty location, if the player can afford. Although this setting is simple, detailed commands (e.g., command per unit) can be easily set up, which bear more resemblance to StarCraft. Similar setting applies to Capture the Flag and Tower Defense. Please check Appendix for detailed descriptions.

**Rewards.** For Mini-RTS, the agent only receives a reward when the game ends ($\pm 1$ for win/loss). An average game of Mini-RTS lasts for around 4000 ticks, which results in 80 decisions for a frame skip of 50, showing that the game is indeed delayed in reward. For Capturing the Flag, we give intermediate rewards when the flag moves towards player's own base (one score when the flag "touches down"). In Tower Defense, intermediate penalty is given if enemy units are leaked.

### 4.2.1 A3C baseline

Next, we describe our baselines and their variants. Note that while we refer to these as baseline, we are the first to demonstrate end-to-end trained AIs for real-time strategy (RTS) games with partial information. For all games, we randomize the initial game states for more diverse experience and

| Frameskip | SIMPLE | HIT_N_RUN |
|---|---|---|
| 50 | 68.4(±4.3) | 63.6(±7.9) |
| 20 | 61.4(±5.8) | 55.4(±4.7) |
| 10 | 52.8(±2.4) | 51.1(±5.0) |

| | Capture Flag | Tower Defense |
|---|---|---|
| Random | 0.7 (± 0.9) | 36.3 (± 0.3) |
| Trained AI | 59.9 (± 7.4) | 91.0 (± 7.6) |

Table 3: Win rate of A3C models competing with built-in AIs over 10k games. **Left:** Mini-RTS. Frame skip of the trained AI is 50. **Right:** For Capture the Flag, frame skip of trained AI is 10, while the opponent is 50. For Tower Defense the frame skip of trained AI is 50, no opponent AI.

| Game | Mini-RTS SIMPLE | | Mini-RTS HIT_N_RUN | |
|---|---|---|---|---|
| | Median | Mean (± std) | Median | Mean (± std) |
| ReLU | 52.8 | 54.7 (± 4.2) | 60.4 | 57.0 (± 6.8) |
| Leaky ReLU | 59.8 | 61.0 (± 2.6) | 60.2 | 60.3 (± 3.3) |
| BN | 61.0 | 64.4 (± 7.4 ) | 55.6 | 57.5 (± 6.8) |
| Leaky ReLU + BN | **72.2** | **68.4** (± 4.3) | **65.5** | **63.6** (± 7.9) |

Table 4: Win rate in % of A3C models using different network architectures. Frame skip of both sides are 50 ticks. The fact that the medians are better than the means shows that different instances of A3C could converge to very different solutions.

use A3C [21] to train AIs to play the full game. We run all experiments 5 times and report mean and standard deviation. We use simple convolutional networks with two heads, one for actions and the other for values. The input features are composed of spatially structured (20-by-20) abstractions of the current game environment with multiple channels. At each (rounded) 2D location, the type and hit point of the unit at that location is quantized and written to their corresponding channels. For Mini-RTS, we also add an additional constant channel filled with current resource of the player. The input feature only contains the units within the sight of one player, respecting the properties of fog-of-war. For Capture the Flag, immediate action is required at specific situations (e.g., when the opponent just gets the flag) and A3C does not give good performance. Therefore we use frame skip 10 for trained AI and 50 for the opponent to give trained AI a bit advantage. All models are trained from scratch with curriculum training (Sec. 4.2.2).

Note that there are several factors affecting the AI performance.

**Frame-skip**. A frame skip of 50 means that the AI acts every 50 ticks, etc. Against an opponent with low frame skip (fast-acting), A3C's performance is generally lower (Fig. 3). When the opponent has high frame skip (e.g., 50 ticks), the trained agent is able to find a strategy that exploits the long-delayed nature of the opponent. For example, in Mini-RTS it will send two tanks to the opponent's base. When one tank is destroyed, the opponent does not attack the other tank until the next 50-divisible tick comes. Interestingly, the trained model could be adaptive to different frame-rates and learn to develop different strategies for faster acting opponents. For Capture the Flag, the trained bot learns to win 60% over built-in AI, with an advantage in frame skip. For even frame skip, trained AI performance is low.

**Network Architectures**. Since the input is sparse and heterogeneous, we experiment on CNN architectures with Batch Normalization [11] and Leaky ReLU [18]. BatchNorm stabilizes the gradient flow by normalizing the outputs of each filter. Leaky ReLU preserves the signal of negative linear responses, which is important in scenarios when the input features are sparse. Tbl. 4 shows that these two modifications both improve and stabilize the performance. Furthermore, they are complimentary to each other when combined.

**History length**. History length $T$ affects the convergence speed, as well as the final performance of A3C (Fig. 5). While Vanilla A3C [21] uses $T = 5$ for Atari games, the reward in Mini-RTS is more delayed ($\sim 80$ actions before a reward). In this case, the $T$-step estimation of reward $R_1 = \sum_{t=1}^{T} \gamma^{t-1} r_t + \gamma^T V(s_T)$ used in A3C does not yield a good estimation of the true reward if $V(s_T)$ is inaccurate, in particular for small $T$. For other experiments we use $T = 6$.

**Interesting behaviors** The trained AI learns to act promptly and use sophisticated strategies (Fig. 6). Multiple videos are available in https://github.com/facebookresearch/ELF.

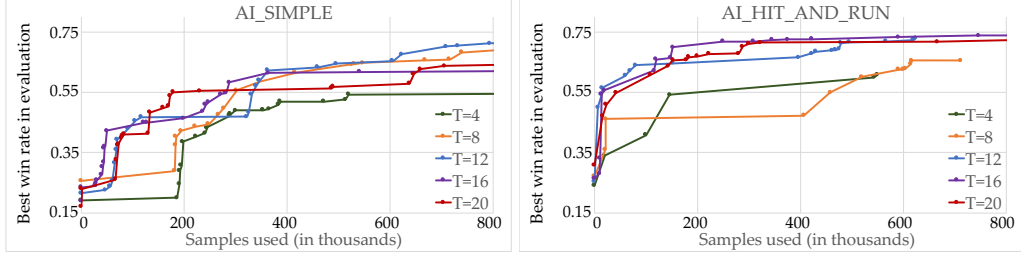

Figure 5: Win rate in Mini-RTS with respect to the amount of experience at different steps $T$ in A3C. Note that one sample (with history) in $T = 2$ is equivalent to two samples in $T = 1$. Longer $T$ shows superior performance to small step counterparts, even if their samples are more expensive.

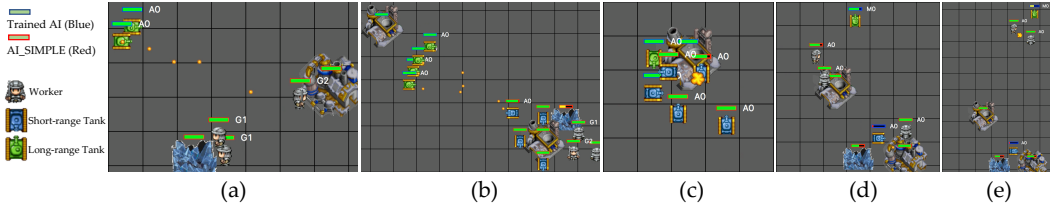

Figure 6: Game screenshots between trained AI (blue) and built-in SIMPLE (red). Player colors are shown on the boundary of hit point gauges. **(a)** Trained AI rushes opponent using early advantage. **(b)** Trained AI attacks one opponent unit at a time. **(c)** Trained AI defends enemy invasion by blocking their ways. **(d)-(e)** Trained AI uses one long-range attacker (top) to distract enemy units and one melee attacker to attack enemy's base.

### 4.2.2 Curriculum Training

We find that curriculum training plays an important role in training AIs. All AIs shown in Tbl. 3 and Tbl. 4 are trained with curriculum training. For Mini-RTS, we let the built-in AI play the first $k$ ticks, where $k \sim \text{Uniform}(0, 1000)$, then switch to the AI to be trained. This (1) reduces the difficulty of the game initially and (2) gives diverse situations for training to avoid local minima. During training, the aid of the built-in AIs is gradually reduced until no aid is given. All reported win rates are obtained by running the trained agents alone with greedy policy.

We list the comparison with and without curriculum training in Tbl. 6. It is clear that the performance improves with curriculum training. Similarly, when fine-tuning models pre-trained with one type of opponent towards a mixture of opponents (e.g., 50%SIMPLE + 50%HIT_N_RUN), curriculum training is critical for better performance (Tbl. 5). Tbl. 5 shows that AIs trained with one built-in AI cannot do very well against another built-in AI in the same game. This demonstrates that training with diverse agents is important for training AIs with low-exploitability.

### 4.2.3 Monte-Carlo Tree Search

Monte-Carlo Tree Search (MCTS) can be used for planning when complete information about the game is known. This includes the complete state $s$ without fog-of-war, and the precise forward model $s' = s'(s, a)$. Rooted at the current game state, MCTS builds a game tree that is biased

| Game | Mini-RTS | | |
|---|---|---|---|
| | SIMPLE | HIT_N_RUN | Combined |
| SIMPLE | **68.4** ($\pm 4.3$) | 26.6($\pm 7.6$) | 47.5($\pm 5.1$) |
| HIT_N_RUN | 34.6($\pm 13.1$) | **63.6** ($\pm 7.9$) | 49.1($\pm 10.5$) |
| Combined(No curriculum) | 49.4($\pm 10.0$) | 46.0($\pm 15.3$) | 47.7($\pm 11.0$) |
| Combined | 51.8($\pm 10.6$) | 54.7($\pm 11.2$) | **53.2**($\pm 8.5$) |

Table 5: Training with a specific/combined AIs. Frame skip of both sides is 50. When against combined AIs (50%SIMPLE + 50%HIT_N_RUN), curriculum training is particularly important.

| Game | Mini-RTS SIMPLE | Mini-RTS HIT_N_RUN | Capture the Flag |
|---|---|---|---|
| no curriculum training | 66.0(±2.4) | 54.4(±15.9) | 54.2(±20.0) |
| with curriculum training | **68.4** (±4.3) | **63.6** (±7.9) | **59.9** (±7.4) |

Table 6: Win rate of A3C models with and without curriculum training. Mini-RTS: Frame skip of both sides are 50 ticks. Capture the Flag: Frame skip of trained AI is 10, while the opponent is 50. The standard deviation of win rates are large due to instability of A3C training. For example in Capture the Flag, highest win rate reaches 70% while lowest win rate is only 27%.

| Game | Mini-RTS SIMPLE | Mini-RTS HIT_N_RUN |
|---|---|---|
| Random | 24.2(±3.9) | 25.9(±0.6) |
| MCTS | 73.2(±0.6) | 62.7(±2.0) |

Table 7: Win rate using MCTS over 1000 games. Both players use a frameskip of 50.

towards paths with high win rate. Leaves are expanded with all candidate moves and the win rate estimation is computed by random self-play until the game ends. We use 8 threads, each with 100 rollouts. We use root parallelization [9] in which each thread independently expands a tree, and are combined to get the most visited action. As shown in Tbl. 7, MCTS achieves a comparable win rate to models trained with RL. Note that the win rates of the two methods are not directly comparable, since RL methods have no knowledge of game dynamics, and its state knowledge is reduced by the limits introduced by the fog-of-war. Also, MCTS runs much slower (2-3sec per move) than the trained RL AI ($\leq$ 1msec per move).

## 5 Conclusion and Future Work

In this paper, we propose ELF, a research-oriented platform for concurrent game simulation which offers an extensive set of game play options, a lightweight game simulator, and a flexible environment. Based on ELF, we build a RTS game engine and three initial environments (Mini-RTS, Capture the Flag and Tower Defense) that run 40KFPS per core on a laptop. As a result, a full-game bot in these games can be trained end-to-end in one day using a single machine. In addition to the platform, we provide throughput benchmarks of ELF, and extensive baseline results using state-of-the-art RL methods (e.g, A3C [21]) on Mini-RTS and show interesting learnt behaviors.

ELF opens up many possibilities for future research. With this lightweight and flexible platform, RL methods on RTS games can be explored in an efficient way, including forward modeling, hierarchical RL, planning under uncertainty, RL with complicated action space, and so on. Furthermore, the exploration can be done with an affordable amount of resources. As future work, we will continue improving the platform and build a library of maps and bots to compete with.

## Footnotes

[1]The GIL in Python forbids simultaneous interpretations of multiple statements even on multi-core CPUs.

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
