[Supplementary Material]

# ELF: An Extensive, Lightweight and Flexible Research Platform for Real-time Strategy Games

November 15, 2017

## 1 Detailed descriptions of RTS engine and games

### 1.1 Overview

On ELF, we thus build three different environments, Mini-RTS, Capture the Flag and Tower Defense. Tbl. 1 shows their characteristics.

Figure 1: Overview of Mini-RTS. **(a)** Tick-driven system. **(b)** Visualization of game play. **(c)** Command system.

| Game Name | Descriptions | Avg game length |
|---|---|---|
| Mini-RTS | Gather resource/build troops to destroy enemy's base. | 1000-6000 ticks |
| Capture the Flag | Capture the flag and bring it to your own base | 1000-4000 ticks |
| Tower Defence | Builds defensive towers to block enemy invasion. | 1000-2000 ticks |

Table 1: Short descriptions of three different environments built from our RTS engine.

### 1.2 Hierarchical Commands

The command level in our RTS engine is hierarchical (Fig. 2). A high-level command can issue other commands at the same tick during execution, which are then executed and can potential issues other commands as well. A command can also issue subsequent commands for future ticks. Two kinds of commands exist, *durative* and *immediate*. Durative commands (e.g., Move, Attack) last for many ticks until completion (e.g., enemy down), while immediate commands take effect at the current tick.

Figure 2: Hierarchical command system in our RTS engine. Top-level commands can issue strategic level commands, which in terms can issue durative and immediate commands to each unit (e.g., ALL_ATTACK can issue ATTACK command to all units of our side). For a unit, durative commands usually last for a few ticks until the goal is achieved (e.g., enemy down). At each tick, the durative command can issue other durative ones, or immediate commands which takes effects by changing the game situation at the current tick.

## 1.3  Units and Game Dynamics

**Mini-RTS**. Tbl. 2 shows available units for Mini-RTS, which captures all basic dynamics of RTS Games: Gathering, Building facilities, Building different kinds of troops, Defending opponent's attacks and/or Invading opponent's base. For troops, there are melee units with high hit point, high attack points but low moving speed, and agile units with low hit point, long attack range but fast moving speed. Tbl. 3 shows available units for Capture the Flag.

Note that our framework is extensive and adding more units is easy.

| Unit name | Description |
|---|---|
| BASE | Building that can build workers and collect resources. |
| RESOURCE | Resource unit that contains 1000 minerals. |
| WORKER | Worker who can build barracks and gather resource. Low movement speed and low attack damage. |
| BARRACKS | Building that can build melee_attacker and range_attacker. |
| MELEE_ATTACKER | Tank with high HP, medium movement speed, short attack range, high attack damage. |
| RANGE_ATTACKER | Tank with low HP, high movement speed, long attack range and medium attack damage. |

Table 2: Available units in Mini-RTS.

| Unit name | Description |
|---|---|
| BASE | Building that can produce athletes. |
| FLAG | Carry the flag to base to score a point. |
| ATHLETE | Unit with attack damage and can carry a flag. Moves slowly with a flag. |

Table 3: Available units in Capture the Flag.

**Capture the Flag**. During the game, the player will try to bring the flag back to his own base. The flag will appear in the middle of the map. The athlete can carry a flag or fight each other. When carrying a flag, an athlete has reduced movement speed. Upon death, it will drop the flag if it is carrying one, and will respawn automatically at base after a certain period of time. Once a flag is brought to a player's base, the player scores a point and the flag is returned to the middle of the map. The first player to score 5 points wins.

**Tower Defense**. During the game, the player will defend his base at top-left corner. Every 200 ticks, increasing number of enemy attackers will spawn at lower-right corner of the map, and travel towards player's base through a maze. The player can build towers along the way to prevent enemy from reaching the target. For every 5 enemies killed, the player can build a new tower. The player will lose if 10 enemies reach his base, and will win if he can survive 10 waves of attacks.

## 1.4   Others

**Game Balance.** We test the game balance of Mini-RTS and Capture the Flag. We put the same AI to combat each other. In Mini-RTS the win rate for player 0 is $50.0(\pm3.0)$ and In Capture the Flag the win rate for player 0 is $49.9(\pm1.1)$.

   **Replay.** We offer serialization of replay and state snapshot at arbitrary ticks, which is more flexible than many commercial games.

# 2   Detailed explanation of the experiments

Tbl. 4 shows the discrete action space for Mini-RTS and Capture the Flag used in the experiments.

   **Randomness**. All games based on RTS engine are deterministic. However, modern RL methods require the experience to be diverse to explore the game state space more efficiently. When we train AIs for Mini-RTS, we add randomness by randomly placing resources and bases, and by randomly adding units and buildings when the game starts. For Capture the Flag, all athletes have random starting position, and the flag appears in a random place with equal distances to both player's bases.

## 2.1   Rule based AIs for Mini-RTS

**Simple AI** This AI builds 3 workers and ask them to gather resources, then builds a barrack if resource permits, and then starts to build melee attackers. Once he has 5 melee attackers, all 5 attackers will attack opponent's base.

   **Hit & Run AI** This AI builds 3 workers and ask them to gather resources, then builds a barrack if resource permits, and then starts to build range attackers. Once he has 2 range attackers, the range attackers will move towards opponent's base and attack enemy troops in range. If enemy counterattacks, the range attackers will hit and run.

## 2.2   Rule based AIs for Capture the Flag

**Simple AI** This AI will try to get flag if flag is not occupied. If one of the athlete gets the flag, he will escort the flag back to base, while other athletes defend opponent's attack. If an opponent athlete carries the flag, all athletes will attack the flag carrier.

| Command name | Description |
|---|---|
| IDLE | Do nothing. |
| BUILD_WORKER | If the base is idle, build a worker. |
| BUILD_BARRACK | Move a worker (gathering or idle) to an empty place and build a barrack. |
| BUILD_MELEE_ATTACKER | If we have an idle barrack, build an melee attacker. |
| BUILD_RANGE_ATTACKER | If we have an idle barrack, build an range attacker. |
| HIT_AND_RUN | If we have range attackers, move towards opponent base and attack. Take advantage of their long attack range and high movement speed to hit and run if enemy counter-attack. |
| ATTACK | All melee and range attackers attack the opponent's base. |
| ATTACK_IN_RANGE | All melee and range attackers attack enemies in sight. |
| ALL_DEFEND | All troops attack enemy troops near the base and resource. |

Table 4: Action space used in our trained AI. There are 9 strategic hard-coded global commands. Note that all building commands will be automatically cancelled when the resource is insufficient.

| Command name | Description |
|---|---|
| IDLE | Do nothing. |
| GET_FLAG | All athletes move towards the flag and capture the flag. |
| ESCORT_FLAG | Move the athlete with the flag back to base. |
| ATTACK | Attack the opponent athlete with the flag. |
| DEFEND | Attack the opponent who is attacking you. |

Table 5: Action space used in Capture the Flag trained AI.