[Reviews · NeurIPS 2017]

Reviewer 1



The main proposal of the paper is a real-time strategy simulator specifically designed for reinforcement learning purposes. The paper presents with several details the architecture of the simulator, along with how gaming is done on it and some experimentations with the software with some RL techniques implemented in the software. Although I think there are good values in making with software for research, I don’t think that NIPS is the right forum for presenting technical papers on them. Machine Learning Open Source Software (MLOSS) track from JMLR or relevant workshop are much relevant for that. And in the current case, a publication in the IEEE Computational Intelligence and Games (IEEE-CIG) conference might be a much better fit. That being said, I am also quite unsure of the high relevance of real-time strategies (RTS) games for demonstrating reinforcement learning (RL). The use of RTS for benchmarking AI is not new, it has been done several time in the past. I am not convinced it will the next Atari or Go. It lacks the simplicity and elegance of Go, while not being particularly useful for practical applications. The proposed implementation is sure designed to allow fast during simulation, making learning can be done much faster (assuming that the bottleneck was the simulator, not the learning algorithms). But such a feat does not justify a NIPS paper per se, there contribution is implementation-wise, not on learning by itself. *** Update following rebuttal phase and comments from area chair *** It seems that the paper still fit in the scope of the conference. I updated my evaluation accordingly, being less harsh in my scoring. However, I maintain my opinion on this paper, I still think RTS for AI is not a new proposal, and I really don't feel it will be the next Atari or Go. The work proposed appears correct in term of implementation, but in term of general usefulness, I still remain to be convinced.

Reviewer 2



ELF is described to be a platform for reinforcement learning research. The authors demonstrate the utility of ELF by implementing a RTS engine with 3 game environments. Further, they show that trained AIs can be competitive under certain scenarios. Curriculum training is shown to be beneficial ELF can be a useful resource to the community.

Reviewer 3



A reinforcement learning platform ELF is introduced in this paper with focus on real time strategy games. The current ELF has game engine supporting three environments, Mini-RTS, Tower Defence and Capture the Flag. It supports efficient parallelism of multiple game instances on multi-core CPU due to multi-thread C++ simulator. ELF can host games written in C/C++. It can switch environment-actor topologies such as one-t-one, one-to-many and many-to-one. Game bot can be trained base on both raw pixels and internal data. Several baseline RL algorithms are implemented in pytorch. Its command hierarchy allows manipulation of commands on different levels. Rule-based AI can also be implemented in ELF. Due to these propoties, they can train game bot for RTS game in end-to-end manner by using ELF, and achieve better performance than built-in rule-based AI. Using ELF, this paper shows that a network with Leaky ReLU and Batch Normalization coupled with long-horizon training and progressive curriculum can beat the rule-based built-in AI. Two algorithms are tested: A3C and MCTS. In A3C, trained AI uses a smaller frame skip than the opponent. These results are helpful to future study. The paper is well-written. The proposed ELF platform largely helps to future research of developing RL algorithm for RTS game. The experiments give some insights about network architecture and training strategy. Although Mini-RTS can be seem as a simplified version of StarCraft, it is interesting to see how ELF and the presented curriculum learning performs on more complex RTS games.